# Dickkopf-2 (DKK2) as Context Dependent Factor in Patients with Esophageal Adenocarcinoma

**DOI:** 10.3390/cancers12020451

**Published:** 2020-02-14

**Authors:** Lars M. Schiffmann, Heike Loeser, Anne Sophie Jacob, Martin Maus, Hans Fuchs, Yue Zhao, Lars Tharun, Ahlem Essakly, Alexander Iannos Damanakis, Thomas Zander, Reinhard Büttner, Wolfgang Schröder, Christiane Bruns, Alexander Quaas, Florian Gebauer

**Affiliations:** 1Department of General, Visceral and Cancer Surgery, University of Cologne, Kerpener Strasse 62, 50937 Cologne, Germany; lars.schiffmann@uk-koeln.de (L.M.S.); annesophiejacob@msn.com (A.S.J.); mauman@gmx.de (M.M.); Hans.fuchs@uk-koeln.de (H.F.); yue.zhao@uk-koeln.de (Y.Z.); alexander.damanakis@uk-koeln.de (A.I.D.); wolfgang.schroeder@uk-koeln.de (W.S.); christiane.bruns@uk-koeln.de (C.B.); 2Department of Pathology, University of Cologne, Kerpener Strasse 62, 50937 Cologne, Germany; heike.loeser@uk-koeln.de (H.L.); lars.tharun@uk-koeln.de (L.T.); Ahlem.Essakly@uk-koeln.de (A.E.); reinhard.buettner@uk-koeln.de (R.B.); Alexander.Quaas@uk-koeln.de (A.Q.); 3Department I of Internal Medicine, Center for Integrated Oncology Aachen Bonn Cologne Duesseldorf, Gastrointestinal Cancer Group Cologne (GCGC), University of Cologne, Kerpener Strasse 62, 50937 Cologne, Germany; Thomas.zander@uk-koeln.de

**Keywords:** response prediction, DKK2, GATA6, EAC

## Abstract

Dickkopf-2 (DKK2) has been described as Wnt/beta-catenin pathway antagonist and its expression is mediated by micro RNA-221 (miRNA-221). So far, there is only limited data characterizing the role of DKK2 expression in esophageal cancer. A tissue micro array of 192 patients with esophageal adenocarcinoma was analyzed immunohistochemically for DKK2, miRNA-221 expression by RNA scope, and GATA6 amplification by fluorescence in-situ hybridization. The data was correlated with clinical, pathological and molecular data (TP53, HER2, *c-myc*, *GATA*6, *PIK3CA,* and *KRAS* amplifications). DKK2 expression was detectable in 21.7% and miRNA-221 expression in 33.5% of the patients. We observed no correlation between DKK2 or miRNA-221 expression and clinico-pathological data DKK2 expression was correlated with TP53 mutations and amplification of *GATA6*. We did not detect a survival difference in dependence of DKK2 for the total cohort, however, in patients without neoadjuvant treatment DKK2 expression correlated with a prolonged survival (median overall-survival 202 vs. 55 months, *p* = 0.012) which turned opposite in patients that underwent neoadjuvant treatment. High amounts of miRNA-221 were in trend associated with a prolonged overall-survival (*p* = 0.070). DKK2 as a Wnt antagonist is associated with prolonged survival in patients without neoadjuvant treatment and changes its prognostic value to the contrary in patients after neoadjuvant therapy. The modulatory effects of neoadjuvant treatment in connection with DKK2 expression are not fully understood, but when considering DKK2 as a tumor marker, it is necessary to see it in the context of neoadjuvant therapy.

## 1. Introduction

Esophageal adenocarcinoma (EAC) is one of the most lethal malignancies in the gastrointestinal system and overall-survival is hardly improving despite advances in personalized medicine [1]. Currently, decisions for or against neoadjuvant treatment regimens are based on clinical parameters obtained during staging routines [2]. As treatment response is of immense importance for patient’s overall outcome, ongoing studies focus on optimizing multimodal treatment concepts with the aim to improve response rates and thereby overall survival [2,3]. It would be of similar importance to improve our ability to predict response rates to multimodal EAC therapy concepts as of today overall response rates to neoadjuvant treatment are only around 50% and a significant number of patients receive neoadjuvant therapy, which is associated with significant toxicity effects without any benefit for non-responder [4].

Dickkopf 2 (DKK2) is a member of the Dickkopf family and was identified as a modulator in the Wnt-beta-catenin pathway via binding of the lipoprotein receptor-related protein 5/6 [5,6,7]. The Wnt-pathway latter has been described to be dysregulated in many human diseases including cancer, and its hyperactivation can lead to aberrant cell growth and tumor progression [8]. It was shown that DKK2 expression is modulated by expression of micro RNA-221 (miRNA-221) in esophageal cancer [9]. So far, it has been assumed that DKK2 is a Wnt-antagonist and that, accordingly, inactivation of DKK2 increases Wnt activity with accelerating tumor progression.

The aim of the present study was to assess the impact of miRNA-221 and DKK2 expression in EAC in a large patient cohort with respect to correlation with clinical parameters and overall-survival, pathological and molecular data.

## 2. Material and Patients

### 2.1. Patients and Tumor Samples

Formalin-fixed and paraffin embedded material of 192 patients with esophageal adenocarcinomas that underwent primary surgical resection or resection after neoadjuvant therapy between 1999–2014 at the Department of General, Visceral and Cancer Surgery, University of Cologne, Germany were analyzed. The standard surgical procedure consisted of a transthoracic en-bloc esophagectomy with two-field lymphadenectomy (abdominal and mediastinal lymph nodes), reconstruction was done by formation of a gastric tube with intrathoracic esophagogastrostomy (Ivor-Lewis esophagectomy) [10]. The abdominal phase was predominantly performed as a laparoscopic procedure (hybrid Ivor-Lewis esophagectomy). Technical details of this operation are described elsewhere [11,12,13]. Patients with locally advanced esophageal cancer (cT3) or evidence for loco regional lymph node metastasis in clinical staging received preoperative chemoradiation (5-Fluouracil, cisplatin, 40Gy) or chemotherapy. Follow-up data were available for all patients (Table 1).

Ethical Approval: All procedures followed the national and institutional ethical standards and were in accordance with the relevant version of the Helsinki Declaration. Informed and ethical approved consent from the local ethics committee (13-091) was obtained from all included patients.

Informed Consent: Informed consent was obtained from all included patients.

Single spot tissue micro arrays (TMA) were built for immunohistochemical analyses. TMA construction was performed as previously described [14,15]. In brief, tissue cylinders with a diameter of 1.2 mm each were punched from selected tumor tissue blocks using a self-constructed semi-automated precision instrument and embedded in empty recipient paraffin blocks. Four μm sections of the resulting TMA blocks were transferred to an adhesive coated slide system (Instrumedics Inc., Hackensack, NJ, USA) for immunohistochemistry.

Expression of DKK2 was correlated with molecular profiles of EAC including analysis of TP53, Her2, *c-myc, GATA6, PIK3CA* mutations and *KRAS* amplification.

### 2.2. Immunohistochemistry for DKK2

Immunohistochemistry (IHC) was performed on TMA slides using the DKK2 rabbit IgG polyclonal antibody (ab38594; dilution 1:200; Abcam, Cambridge, UK). All immunohistochemical stainings were performed using the Leica BOND-MAX stainer (Leica Biosystems, Wetzlar, Germany) according to the protocol of the manufacturer.

The TMA was scored manual by two pathologists (A.Q. and H.L.) according to a 4-tier-scoring system. We defined Score 3+ as a strong staining of ≥30% of tumor cells or moderate staining ≥70%. Score 2+ was defined as weak staining in >70% or moderate staining in >30 and ≤70% or as strong staining in ≤30% of tumor cells. Score 1+ was assigned when ≤70% of tumor cells were weakly positive or ≤30% were moderately stained. Less staining was defined as negative (Score 0). Discrepant results were resolved by consensus review

### 2.3. RNAscope for miRNA221

The RNAscope assay was performed according to manufacturer’s instruction.

In brief, paraffin-embedded TMA blocks were cut in 5 μm sections, pretreated according to an extended protocol (30 min for pretreatment 2 and 3), digested and hybridized at 40 °C in the HybEZ oven with human miRNA-221 mRNA probe provided by *Advanced Cell Diagnostics Europe*. Incubation time with Hematoxylin was 10 s.

Target expression was compared to both negative (dapB) and positive (PPIB) controls. Scoring of signals was done as recommend by the manufacturer with no staining or less than one molecule per 10 cells = score 0; 1–3 dots/cell = score 1; 4–9 dots/cell = score 2; 10–15 dots/cell = score 3 and >15 dots/cell = score 4. DapB score was 0 and PPIB score was 2. Positivity was defined as a score > 0.

The evaluation of immunohistochemical expression and RNAscope was assessed manually by two pathologists (AQ and HL). Discrepant results, which occurred only in a small number of samples, were resolved by consensus review.

### 2.4. Fluorescence In-Situ Hybridization (FISH)

Fluorescence in-situ hybridization (FISH) analysis for evaluation of *GATA6* gene copy numbers was performed with GATA6-20-GR Probe (Empire Genomics, New York, NY, USA) and the Zytolight centromere 18 (CEN18) Probe (Zytovision Bremerhaven, Germany). Three μm tissue sections on slides (SuperFrost Plus) were mounted by heating, followed by deparaffinization, protease digestion and washing steps (VP2000 processor system, Abbott Molecular, Wiesbaden, Germany) and hybridization at 37 °C overnight with the FISH Probe. The slides were stained with DAPI before analysis. Cases were further evaluated only when normal tissue nuclei displayed one or two clearly distinct signals of green *GATA6* and orange *CEN18*. The reading strategy followed that of the *c-myc*-FISH probe to evaluate areas of cluster amplification. Tumor tissue was scanned for amplification hot spots of *GATA6* signals using ×63 objective (DM5500 fluorescent microscope; Leica).

### 2.5. Genomic Data Analysis

Genomic data were analyzed by cBioPortal (cbioportal.org) and plotted for visualization of interactions between DKK2 and GATA6 [16,17].

Underlying genetic data were extracted from the TCGA dataset for esophageal adenocarcinoma [18].

### 2.6. Statistical Analysis

Clinical data were collected prospectively according to a standardized protocol. SPSS Statistics for Mac (Version 21, SPSS) was used for statistical analysis. Interdependence between stainings and clinical data were calculated using the chi-squared and Fisher’s exact tests, and displayed by cross-tables. Survival curves were plotted using the Kaplan-Meier method and analyzed using the log-rank test. All tests were two-sided. *p* values < 0.05 were considered statistically significant.

## 3. Results

### 3.1. Patients’ Baseline Characteristics

A total of 175 patients of 192 on the TMA were immunohistochemically interpretable for DKK2 and 176 patients for miRNA-221. Reasons for non-informative cases (16 and 17 spots; 8.3% and 8.9% on the TMA) included lack of tissue samples or absence of unequivocal cancer tissue in the TMA spot. Clinicopathological data is summarized within Table 1. Patients were predominantly men (male n = 175, 91.1%, female n = 17, 8.9%). The median age of the entire patients’ cohort was 65.2 years (range 33.6 - 85.6 years) at time of diagnosis. Neoadjuvant treatment (chemo-or radiochemotherapy) was administered in 142 patients (73.6%) before operation.

### 3.2. miRNA-221 and DKK2 Expression in Esophageal Adenocarcinoma

Expression of DKK2 was detectable in 38 patients (21.7%) and 59 patients (33.5%) for miRNA-221, both showed an intracellular staining pattern (Figure 1 and Figure 2
Figure 1; Figure 2). Amplification of GATA6 was detectable in 18 patients (10.7%) A correlation between clinico-pathological data and DKK2 or miRNA-221 expression could not be revealed by cross-table analysis (Table 1). DKK2 expression was correlated with TP53 wild-type tumors (*p* = 0.034) and *GATA6* amplification (*p* = 0.033) (Figure 3, Table 1). We observed no correlation between Her2 expression or amplification of *c-myc, PIK3CA, KRAS,* and DKK2 expression.

### 3.3. DKK2 Expression Is Associated with Shortened Overall-Survival in Patients After Neoadjuvant Treatment

Observing the entire patient cohort, a significant difference between patients with and without DKK2 expression could not be observed (Figure 4A). In patients without neoadjuvant treatment, there was a survival difference between patients with and without DKK2 expression. Patients with the presence of DKK2 expression showed a median OS of 202 months (95%CI not calculable) and patients without DKK2 expression a median OS of 55.9 months (95%CI 26.1—85.8 months, *p* = 0.012) (Figure 4B). However, in patients receiving neoadjuvant treatment, tumors with DKK2 expression showed a significant shortened OS compared to patients without DKK2 expression. The median OS in patients with DKK2 expression was 26.2 months (95% confidence interval (CI) 13.4–39.2 months) while median OS was 32.8 months (95% CI 13.8–51.3 months, *p* = 0.022) in patients without DKK2 expression (Figure 3C). Expression of miRNA-221 was not correlated with OS in the present patient cohort, however, in patients without neoadjuvant treatment miRNA-221 positive tumors showed a strong trend towards shortened OS (*p* = 0.070) with a median OS of 64.0 months in miRNA-221 negative patients (95%CI not calculable) compared to a median OS of 22.1 month (95%CI 13.3–30.8 months). Amplification of *GATA6* did not serve as prognostic marker, either for the entire patients group or stratified in patients with or without neoadjuvant treatment. The missing statistical relevance could be based on a rather small number of positive patients in the group of patients underwent upfront surgery, since none of the patients in this particular group with amplification of *GATA6* died during the follow-up period. In a multivariate cox-regression analysis, DKK2 serves as independent prognostic marker in the neoadjuvant group (*p* = 0.017) with a calculated hazard ratio of 1.895 (95%CI 1.120–3.208) but not in the group of patients after primary surgery (Table 2).

### 3.4. Genomic Data Analysis

Possible interactions between DKK2 and GATA6 were analyzed on genomic level and visualized in Figure 5. Regulation of the Wnt pathway by LRP binding becomes evident. We observed a strong correlation between DKK2 and GATA6 genomic alterations on genomic level supporting the immunohistochemical and FISH data on our own data set.

## 4. Discussion

### 4.1. DKK2 Expression and Its Impact on Survival

In the present study, we focused on the prognostic impact of DKK2 and miRNA-221 expression in EAC by analyzing a large patient collective by TMA technique. DKK2 expression was found in 21.7% of EAC patients, however, there was no correlation between clinical or pathological features (e.g., pT and pN stage) and the expression of DKK2. We found no difference in survival with regard to the DKK2 expression in the entire cohort. However, distinct subgroup analysis revealed that an influence of DKK2 on survival was present upon a multimodal treatment concept. DKK2 expression was associated with a shortened OS in those patients who underwent neoadjuvant treatment while in patients that underwent upfront surgery, DKK2 positive patients had a prolonged OS.

Our results for patients after a primary resection are consistent with previous studies for gastric, colon, breast and hepatocellular carcinoma, where DKK2 was found as a tumor suppressor and its loss is associated with activation of the Wnt-pathway and consecutive tumor progression [19,20,21,22,23,24]. However, this effect is turned opposite in patients after neoadjuvant radiochemotherapy. In these patients, neoadjuvant radiochemotherapy seems to reverse the effects of the tumor suppressor DKK2. Patients after neoadjuvant radiochemotherapy showed a worsened overall survival depending on DKK2 expression than patients without DKK2 detection. As already shown by the study from Xiao et al., the role of DKK2 in cancer and its regulation on the Wnt-pathway seems to be regulated more complexly than previously assumed [25]. For future analysis, association of DKK2 expression and histopathological response in neoadjuvantly treated patients would be of high interest as a potential mechanism behind our findings.

### 4.2. Role of GATA6

The association between DKK2 expression and high *GATA6* levels cannot be elucidated at this point. An overexpression of the transcription factor *GATA6* is associated with an aggressive phenotype in different tumors. Recent studies have shown an influence on the cell cycle in gastric carcinoma, where a depletion of *GATA6* led to a cycle arrest in the M-phase and thus to significantly slower tumor growth. A link to DKK2 has not yet been described and possible synergistic effects remain speculative. A previous study showed a direct influence of *GATA6* on DKK1 and thus on Wnt regulation in pancreatic carcinoma. Binding of *GATA6* to the DKK1 promoter lead to a down-regulation of DKK1 and thus to a consecutive activation of the Wnt pathways and tumor progression [26]. A direct influence of *GATA6* on the Wnt pathway has been described during embryonic heart development. [27]. Whether these mechanisms also apply in malignancies and in esophageal carcinoma in particular, as well as in DKK2, is unknown and needs further investigation.

### 4.3. Impact of MiRNA-221

MiRNA-221-positive, neoadjuvantly treated tumors showed a strong trend towards a shortened OS (*p* = 0.070) supporting previous results where miRNA-221 expression is associated with 5-fluouracil resistance and tumor growth [9]. In the study by Wang and colleagues, DKK2 expression decreased under prolonged stimulation with 5-fluourcil leading to an increased Wnt/b-catenin signaling and consecutive tumor progression. Due to the design of the present study we cannot assess whether DKK2 changes under administration of neoadjvuant treatment. It would be interesting to investigate whether DKK2 expression underlies dynamic changes during neoadjvuant treatment or not. This should be addressed by comparing pretreatment biopsies (before neoadjuvant treatment) with surgical specimens (after neoadjuvant treatment) in the future.

## 5. Conclusions

In summary, our study identifies DKK2 and miRNA-221 expression in neoadjuvant treated EAC as prognostic factors predicting a detrimental prognosis. Though our study collective is quite large, specific subgroup analyses lead to small subgroups at the end. This requires validation in larger prospective cohorts as well as mechanistic studies to clarify the complex context-dependent role of DKK2 in EAC. This should offer safe and reliable future treatment concepts involving this pathway.

## Figures and Tables

**Figure 1 cancers-12-00451-f001:**
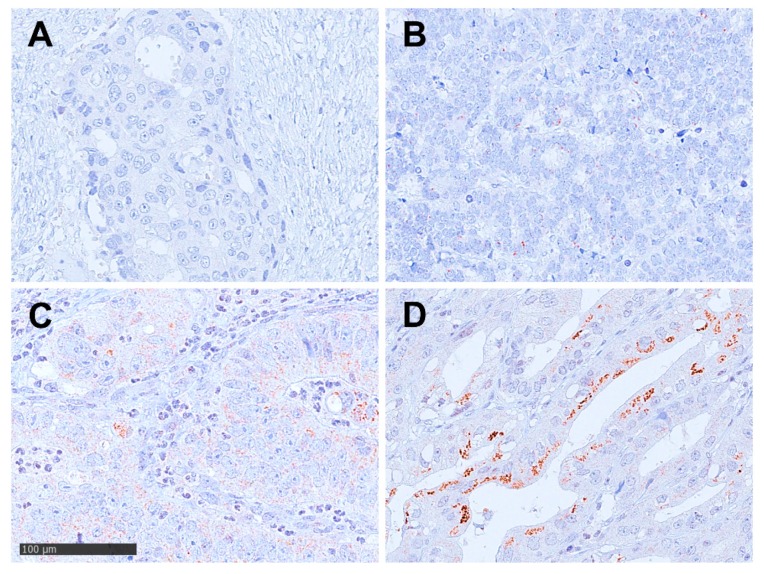
Immunohistochemistry of Dickkopf-2 (DKK2) in esophageal adenocarcinoma according to a 4-tier-scoring system; magnification ×400 (**A**) Negative staining for DKK2 (Score 0); (**B**) weak staining in tumor cells (Score 1+); (**C**) weak to moderate staining in most tumor cells (Score 2+); (**D**) strong staining pattern in >30% of the tumor cells (Score 3+).

**Figure 2 cancers-12-00451-f002:**
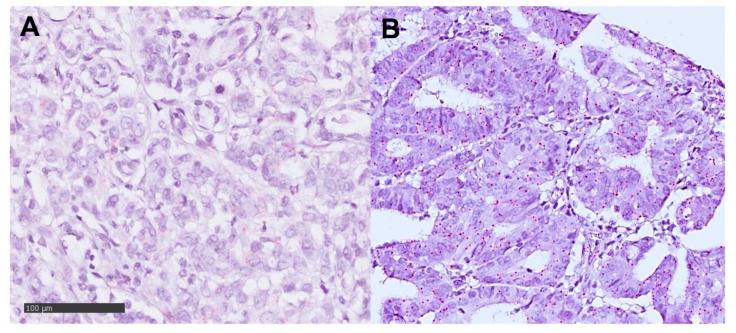
RNA-Scope analysis of mi-RNA221 in esophageal adenocarcinomas; magnification ×400 (**A**) negative tumor cells with less than 1 red signal in 10 tumor cells; (**B**) mi-RNA221 expression in tumor cells (red signals).

**Figure 3 cancers-12-00451-f003:**
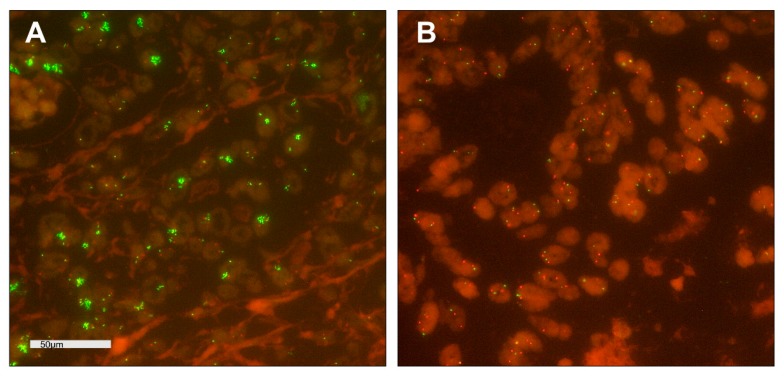
Fluorescence in-situ hybridization (FISH) analysis of *GATA6* in arrayed esophageal adenocarcinomas; magnification ×630; Copy Number Variations; *GATA6* ≙ green; *CEN18* ≙ orange; (**A**) tumor cells with cluster amplification of green *GATA6* signals; (**B**) tumor cells with normal distribution of *GATA6* and *CEN18* signals.

**Figure 4 cancers-12-00451-f004:**
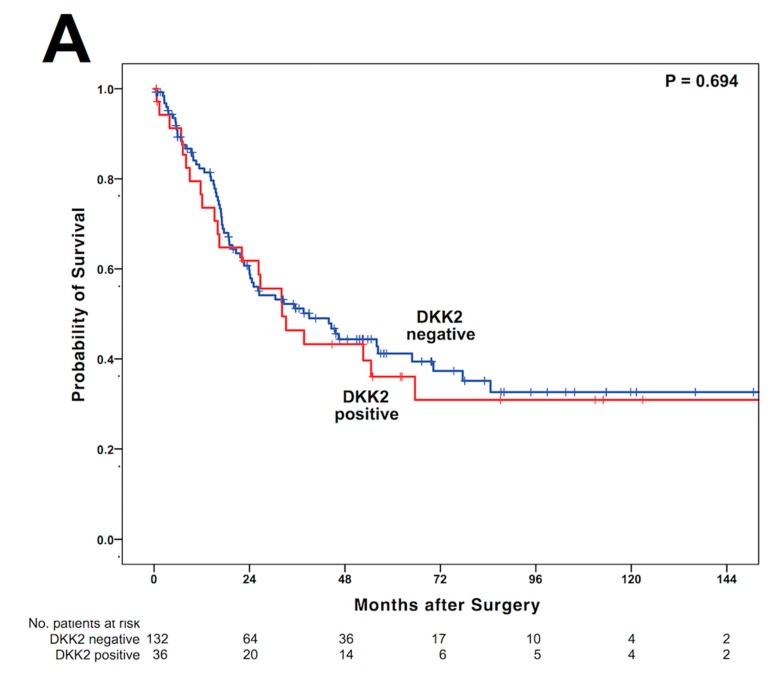
Kaplan–Meier survival analysis and survival plots for DKK2 expression and the entire cohort (**A**), patients after primary surgery without neoadjuvant treatment (**B**), and patients after neoadjuvant treatment (**C**).

**Figure 5 cancers-12-00451-f005:**
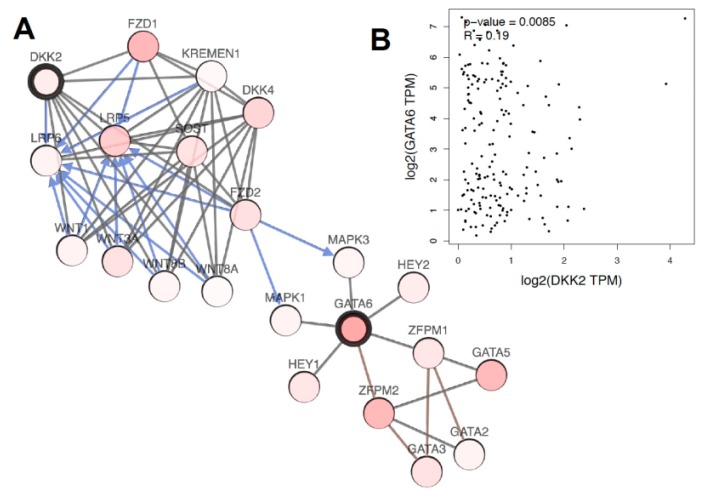
Visualization of the connection of DKK2 and GATA6 on gene level based on TCGA data, analyzed by cBioportal (**A**). Correlation of gene amplification between DKK2 and GATA6 in esophageal adenocarcinoma (**B**).

**Table 1 cancers-12-00451-t001:** Clinico-pathological parameters for the entire patients´ cohort.

Factor				DKK2 Expression					miRNA-221 Expression					GATA 6				
				Negative		Positive		*p* Value	Negative		Positive		*p* Value	Negative		Positive		*p* Value
sex	female	15	8.6%	13	86.7%	2	13.3%		7	50.0%	7	50.0%		12	80.0%	3	20.0%	
	male	160	91.4%	124	77.5%	36	22.5%	0.529	110	67.9%	52	32.1%	0.237	138	90.2%	15	9.8%	0.206
age group	<65 yrs	90	53.6%	69	76.7%	21	23.3%		56	63.6%	32	36.4%		77	88.5%	10	11.5%	
	>65 yrs	78	46.4%	64	82.1%	14	17.9%	0.449	54	68.4%	25	31.6%	0.316	66	89.2%	8	10.8%	1.000
Tumor stage	pT 1	24	13.7%	19	79.2%	5	20.8%		21	75.0%	7	25.0%		18	81.4%	4	18.2%	
	pT 2	10	5.7%	7	70.0	3	30.0%		6	60.0%	4	40.0%		8	80.0%	2	20.0%	
	pT 3	133	76.0%	104	78.2%	29	21.8%		83	63.8%	47	36.2%		116	90.6%	12	9.4%	
	pT 4	8	4.6%	7	87.5%	1	0.6%	0.846	6	85.7%	1	14.3%	0.448	8	100%	0	0%	0.336
Lymph node metastasis	pN 0	70	40.0%	51	72.9%	19	27.1%		50	69.4%	22	30.6%		59	86.8%	9	13.2%	
	pN +	105	60.0%	86	81.9%	19	18.1%	0.818	67	64.4%	37	35.6%	0.563	91	91.0%	9	9.0%	0.266
UICC stage	I	36	20.7%	27	75.0%	9	25.0%		24	68.6%	11	31.4%		30	85.7%	5	14.3%	
	II	35	20.1%	24	68.6%	11	31.4%		23	69.7%	10	30.3%		31	31.2%	3	8.8%	
	III	78	44.8%	64	82.1%	14	17.9%		45	61.6%	28	38.4%		65	89.0%	8	11.0%	
	IV	25	14.4%	21	84.0%	4	16.0%	0.349	16	66.7%	8	33.3%	0.826	23	92.0%	2	8.0%	0.853
neoadjuvant therapy	no	60	34.3%	50	83.3%	10	16.7%		38	66.7%	19	33.3%		55	94.8%	3	5.2%	
	yes	115	65.7%	87	75.7%	28	24.3%	0.334	70	64.2%	39	35.8%	0.864	95	86.4%	15	13.6%	0.118

**Table 2 cancers-12-00451-t002:** Multivariate cox-regression analysis for patients after primary surgery and neoadjuvant treatment.

Factor	Primary Surgery				Neoadjuvant Treatment			
	Hazard Ratio	95% Confidence Interval		*p* Value	Hazard Ratio	95% Confidence Interval		*p* Value
		Lower	Upper			Lower	Upper	
sexmale vs. female	9.623	0.001	.	0.994	3.398	1.048	11.002	0.042
age groups <65 yrs vs. >65 yrs	2.163	0.802	5.832	0.127	1.203	0.724	2.000	0.475
Tumor stage pT1/2 vs. pT3/4	3.585	0.804	15.974	0.094	1.122	0.541	2.327	0.757
lymph node metastasis pN0 vs. pN+	12.063	2.641	55.104	0.001	2.246	1.284	3.929	0.005
DKK2 neg. vs. pos.	0.001	0.001	962.240	0.959	1.895	1.12	3.208	0.017

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
