# Peer review of "Dickkopf-2 (DKK2) as Context Dependent Factor in Patients with Esophageal Adenocarcinoma"

_cancers, 2020, doi:10.3390/cancers12020451_

Round 1

Reviewer 1 Report

The manuscript is much improved and most of my concerns were adequately addressed.

There is one exception:
In response to my inquiry as to DKK in treatment response, the authors simply state in the discussion that there was no relationship. They do not provide any data to support that statement nor is assessment of response described in materials and methods. They should either provide the response assessment or rephrase the discussion statement.

Author Response

We thank the reviewer for his comment, that we initially misunderstood, and we apologize for that. We adopted the revised manuscript accordingly, please compare lines 249-251 in the revised manuscript

Reviewer 2 Report

The authors have improved the quality of their manuscript.

But tables are still hard to see. Please draw a line to distinguish each group to make it easier to see.

Author Response

We followed the reviewers remark and drew lines within the tables.

Reviewer 3 Report

Please change the sentence in line 148 (Underlying genetic data were extracted from the TCGA dataset for esophageal cancer [16]) to the following (Underlying genetic data were extracted from the TCGA dataset for esophageal adenocarcinoma [16]).

The authors have adequately addressed the questioned points.

Author Response

We revised the manuscript as suggested.

This manuscript is a resubmission of an earlier submission. The following is a list of the peer review reports and author responses from that submission.

Round 1

Reviewer 1 Report

The paper “Dickkopf-2 (DKK2) as prognostic factor in patients with esophageal adenocarcinoma in the context of neoadjuvant treatment” is interesting but I have some comments.   

The point should associate the expression of DKK2 in the esophageal cancer patients with the overall survival. In patients without neoadjuvant treatment DKK2 expression correlated with a prolonged survival which turned opposite in patients that underwent neoadjuvant treatment(p = 0.012). However, the author did not describe the impact of DKK2 expression in different population. We can not find the exclusion and exclusion criteria in this article. The auther did not explain the regiment of neoadjuvant chemotherapy. The results can not support the conclusion “ Dickkopf-2 (DKK2) as prognostic factor in patients with esophageal adenocarcinoma in the context of neoadjuvant treatment”.

Reviewer 2 Report

As there is still insufficient data available about the development and molecular characteristics of EAC, this is a highly relevant contribution to clinical research on EAC. However, the work needs to become more thorough before it can be published. Specifically, there are 4 major concerns:

Figure 1 should include separate examples for strong and moderate staining Figure 2: staining is very weak. A better quality picture as well as a negative control should be provided Correlations with genetic characteristics are cursory at best. This is a problem with the GATA6 analysis that is not sufficiently connected with the overall study. If GATA6 data are to be used, the FISH analysis needs to be presented in full and GATA status should be included in table 1. With regard to the impact of DKK2 in primary resected vs neoadjuvantly treated tumors, 2 points remain vague: (1) are there sufficient patients in the primarily resected DKK2+ group to be sure of anything. For this reason figure 3 should include the number of patient per subgroup for easier interpretation. (2) does DKK2 have any impact on therapy response that may have contributed to the result. That needs to be addressed in the discussion.

Minor points:

Both tables need to be reformatted: they are not clearly arranged with linebreaks in many of the cells. It is annoying to have to puzzle over each cell. The discussion should be better structured, so that the different aspects are clearly arranged. As the authors with could cause state the necessity of specific subgroup analysis, they should address the fact that some of the subgroups in this study are very small.

Reviewer 3 Report

This study presents data of DDK2 as prognostic factor in patients with ESC.

Several issuese are presented below

I think figure legend of figure3B and C are opposite. 3B should be patients without neoadjuvant treatments. Tumor samples are surgical specimens. DKK2 expression could change after chemotherapy as mentioned in article. Is there any biopsy samlple before treatment? This is important parts to discuss in patients recieved neoadjuvant treatments. Tables are not clear to see. You should write about sample size of figure 3 in the text or figure legend. Not only in the table.

Reviewer 4 Report

The authors describe DKK2 as a prognostic factor in patients with esophageal adenocarcinoma.

The results indicated that DKK positivity is a positive prognostic factor in patients without neoadjuvant therapy; however, DKK positivity is a negative prognostic factor in patients with neoadjuvant therapy. Although this is an interesting and essential investigation, some revisions might be needed.

Major revision

To explain the opposite impact of DKK on survival according to the administration of neoadjuvant therapy, more specific data is needed to support this phenomenon. Based on the discussion of the manuscript, neoadjuvant therapy decreases the immune cell infiltration in tumors. Does DKK expression correlate to immune cell infiltration after neoadjuvant therapy?   To suggest that DKK is a positive prognostic factor in patients without neoadjuvant therapy, validation cohort will be needed to confirm the prognostic impact of DKK. Does genomic data analysis of TCGA data only include esophageal adenocacrinoma? Original TCGA data included approximately 50% of patients with squamous cell cacrinoma, demanding exclusion of those patients for clarifying the interaction of DKK3 and GATA6 in esophageal adenocacrinoma. If the authors already excluded those patients, please add such manipulation in the method.

Minor revision

If possible, Wnt pathway expression is preferred to be included for confirming suppression of the Wnt pathway in patients with DKK positive tumor.